# Activity of botulinum neurotoxin X and its structure when shielded by a non-toxic non-hemagglutinin protein
Markel Martínez-Carranza [1,6], Jana Škerlová [1,6], Pyung-Gang Lee[2,3,4], Jie Zhang[2,3,4], Ajda Krč [1], Abhishek Sirohiwal [1], Dave Burgin[5], Mark Elliott[5], Jules Philippe[5], Sarah Donald[5], Fraser Hornby[5], Linda Henriksson[1], Geoffrey Masuyer [1], Ville R. I. Kaila [1], Matthew Beard[5], Min Dong [2,3,4] & Pål Stenmark [1]

Botulinum neurotoxins (BoNTs) are the most potent toxins known and are used to treat an increasing number of medical disorders. All BoNTs are naturally co-expressed with a protective partner protein (NTNH) with which they form a 300 kDa complex, to resist acidic and proteolytic attack from the digestive tract. We have previously identified a new botulinum neurotoxin serotype, BoNT/X, that has unique and therapeutically attractive properties. We present the cryo-EM structure of the BoNT/X-NTNH/X complex and the crystal structure of the isolated NTNH protein. Unexpectedly, the BoNT/X complex is stable and protease-resistant at both neutral and acidic pH and disassembles only in alkaline conditions. Using the stabilizing effect of NTNH, we isolated BoNT/X and showed that it has very low potency both in vitro and in vivo. Given the high catalytic activity and translocation efficacy of BoNT/X, low activity of the full toxin is likely due to the receptor-binding domain, which presents very weak ganglioside binding and exposed hydrophobic surfaces.

Botulinum neurotoxins (BoNTs) are a family of bacterial protein toxins produced by *Clostridium botulinum* and other clostridial bacteria. They are the most potent toxins known and cause flaccid paralysis by targeting motor neurons, translocating their catalytic domain into the cytosol, and cleaving SNARE proteins (soluble N-ethylmaleimide-sensitive factor attachment protein receptors) essential for vesicle–membrane fusion. The affected neurons are rendered unable to release neurotransmitters into the neuromuscular junction (NMJ), causing muscle paralysis[1].

BoNTs are 150 kDa proteins and consist of three domains: the 50 kDa catalytically active light chain (LC) and the 100 kDa heavy chain that is further divided into the translocation domain (H$_N$) and the receptor-binding domain (H$_C$). Structurally, the LC and translocation domain form one closely connected unit, referred to as LH$_N$. Canonical BoNTs are classified into seven traditional serotypes, alphabetically named BoNT/A-G[2]. There are also mosaic toxins composed of domains of different serotypes. A novel serotype named BoNT/X was identified in the genome of *C. botulinum* strain 111, which also contains a BoNT/B gene on a plasmid and was originally isolated from an infant botulism case in Japan[3–5]. The BoNT/X cluster in *C. botulinum* strain 111 is of the OrfX-type and encodes an

additional OrfX2 protein named OrfX2b[3]. BoNT/X has emerged as a promising vehicle for intracellular delivery of therapeutic nanobodies, a canvas for engineering substrate specificity, and a potential way to target novel medical conditions[6–8]. BoNT/X has low sequence similarity to other BoNTs (30% sequence identity) and is not recognized by antisera against other known serotypes. It also exhibits a unique substrate specificity as it cleaves VAMP1/2/3 at a distinct site from other BoNTs, and is the only BoNT capable of cleaving the non-canonical SNAREs VAMP4, VAMP5, and Ykt6.

The only structural information available for BoNT/X and its corresponding NTNH to date is the crystal structure of the toxin's highly active LC[9].

Recently, several "botulinum neurotoxin-like toxins" have been discovered. These toxins (BoNT/X, BoNT/En, and PMP1) are closely related to the established BoNTs but form a distinct evolutionary branch[3,10,11]. PMP1 is the first toxin in this family to target insects, specifically the malaria vector *Anopheles* mosquitos[11]. It is not known if BoNT/X or BoNT/En target insects, it should, however, be noted that these toxins were found in bacterial strains isolated from the human gut or from animal feces, while PMP1 was identified in a mosquitocidal strain isolated from mangrove soil.

[1]Department of Biochemistry and Biophysics, Stockholm University, Stockholm, Sweden. [2]Department of Urology, Boston Children's Hospital, Boston, MA, USA. [3]Department of Microbiology, Harvard Medical School, Boston, MA, USA. [4]Department of Surgery, Harvard Medical School, Boston, MA, USA. [5]Ipsen Bioinnovation, Abingdon, UK. [6]These authors contributed equally: Markel Martínez-Carranza, Jana Škerlová. ✉e-mail: min.dong@childrens.harvard.edu; stenmark@dbb.su.se

The disease botulism is most commonly contracted as a foodborne illness when the patient ingests poorly conserved food where toxigenic bacteria have grown and produced the neurotoxin. In order to reach the neurons, BoNTs need to resist the acidic environment and proteases present in the gastrointestinal (GI) tract of the host. Several non-toxic neurotoxin-associated proteins encoded in the *bont* gene clusters are known to protect BoNT by forming high molecular weight assemblies[12]. All known *bont* genes are neighbored in their gene clusters by an *ntnh* gene (nontoxic non-hemagglutinin protein). Together BoNT and NTNH proteins form the 300 kDa minimal progenitor toxin complex (M-PTC). So far the crystal structures of BoNT/A and BoNT/E in complex with their respective NTNH proteins have been solved[13,14], revealing a tight complex where NTNH shares the same general fold as BoNT. NTNH is likely a result of a gene duplication event and has later lost the receptor-binding ability and proteolytic activity[15]. The BoNT-NTNH complexes are held together by several pH-dependent contacts that release the neurotoxin once the complex leaves the acidic GI tract[16], and greatly enhance the oral potency of the neurotoxins by 10 to 20-fold compared to BoNT alone[17].

BoNT/X holds valuable potential in translational medicine. Since BoNT/X displays a unique profile of SNARE protein targets, the catalytic domain is a novel potential tool to modulate cell secretion. Furthermore, a construct encompassing the BoNT/X LC and translocation domain (LH$_N$) has been shown to more efficiently translocate its LC through the endosomal membrane into the cytosol compared to analogous fragments of BoNT/A and BoNT/B[8]. Exploiting this feature, BoNT/X chimeras have successfully been used for delivery of single-domain antibodies into neurons, as well as Cas13 and Cas9[6,8]. Additionally, the unique specificity of LC/X has effectively been altered using an innovative phage-assisted evolution method, breaking new ground for the possible uses of the toxins both as therapeutics and as research tools[7].

Here we have studied the potency of isolated recombinant BoNT/X, determined the structure of the 300 kDa BoNT/X-NTNH/X complex, and investigated the complex's pH-dependent stability. The structure of BoNT/X suggests it is a functional toxin despite its weak activity on the mammalian models tested so far. It provides a template for the design and refinement of BoNT/X-derived biotechnological tools for intracellular delivery of therapeutic molecules or targeted secretion inhibitors[7,8,18].

## Results and discussion
### Production of active BoNT/X

Full-length BoNT/X, and especially the isolated binding domain of BoNT/X (Hc/X), are difficult to express and purify with good yields and purity[3]. Here, we utilized the stabilizing effect of NTNH on BoNT/X to produce full-length active toxin. We co-expressed BoNT/X and its corresponding NTNH, which allowed us to successfully produce both the recombinant 300 kDa M-PTC complex for structural determination, and the active 150 kDa BoNT/X for in vitro and in vivo activity studies (Supplementary Fig. S1).

For safety reasons, active BoNT/X was cloned from two different fragments, one spanning the heavy chain and another one spanning the BoNT/X LC with the activation loop of BoNT/C. This strategy allows for a more efficient and controllable activation of the toxin using Factor Xa, as described by Ipsen (patent WO2020065336A1). This construct was then co-expressed with NTNH/X to facilitate a more accurate characterization of this new serotype's activity.

### Activity of BoNT/X in cultured neurons

First, the ability of BoNT/X to reach and cleave intracellular VAMP was assessed in rat cortical neurons (Fig. 1). Proteolytic degradation was observed for VAMP2 and VAMP4, with 10 nM BoNT/X able to cleave 58% of both substrates (Fig. 1 and Supplementary Fig. S2). Although this activity shows BoNT/X is functional on neurons, it appears considerably less potent than other serotypes. For comparison, BoNT/B presented a sub-picomolar EC$_{50}$ when analyzing VAMP2 cleavage in rat spinal cord neurons[19], which was similar to BoNT/A cleavage of SNAP25 in rat cortical neurons[20]. Remarkably, BoNT/X was only marginally more efficient

than LH$_N$/X, a fragment that lacks the receptor-binding domain, and was able to degrade 42% of intracellular VAMP2. Previous reports evaluating the functionality of LH$_N$ showed that it could retain some level of activity on cultured neurons, but was approximately 10$^5$-fold less potent than the full-length BoNT/A and B toxins[19,21]. The activity observed here with LH$_N$/X suggests this fragment also has the intrinsic ability to transport LC/X inside neuronal cells, within a similar range to what was observed for LH$_N$/B, which presented an EC$_{50}$ of 15 nM for VAMP2 cleavage. This is consistent with previous observations where association of LH$_N$/X with the binding domain of BoNT/A formed a chimeric toxin platform for intracellular delivery of cargo proteins[8].

### Activity of BoNT/X in animal models

In the original characterization of BoNT/X[3], no active holotoxin was produced due to safety considerations. However, an initial assessment of its toxicity was performed by synthesizing a limited amount of complete toxin using sortase-mediated ligation of the isolated functional LH$_N$ and H$_C$ domains. Although H$_C$/X was observed to be relatively unstable on its own, which led to a low yield of ligated toxin, the product from the sortase reaction caused local flaccid paralysis in mice Digit Abduction Score (DAS) assays, when administered at high concentrations. Here we analyzed the activity of full-length BoNT/X holotoxin (not sortase ligated) utilizing both the mouse phrenic nerve hemidiaphragm (mPNHD) assay and the DAS assay.

The mPNHD assay provides a useful ex vivo assessment on the effects of BoNTs by measuring a decrease in the contraction amplitude of the indirectly stimulated muscle[22]. BoNT/X was directly compared to BoNT/A and BoNT/B. As expected, the two controls, 10 pM BoNT/A and BoNT/B produced a time-dependent paralysis of the hemidiaphragm muscle ($t_{50}$ = 39 ± 0.8 min, and 47 ± 1.5 min ($n$ = 3), respectively). In contrast, an equivalent dose of BoNT/X (10 pM, $n$ = 3) had no effect on diaphragm muscle contraction, nor did LH$_N$X at a higher dose (100 pM) (Fig. 1c), thus suggesting that the low potency observed in vitro is too weak to translate into animal toxicity.

Next, BoNT/X holotoxin was evaluated in the DAS assay, which is commonly used to measure the paralytic effects of BoNTs on muscles in vivo[23]. Injections of 1 or 2 μg of BoNT/X in mice did not cause any paralysis (Fig. 1d and Supplementary Tables 1 and 2). This shows the lack of toxicity of BoNT/X since typical doses for BoNT/A and /B are normally within the pM range to observe local paralysis of the leg muscle. In addition, the mice's body weight was not affected by injection of 1 μg of BoNT/X (Supplementary Table 3), suggesting that the toxin does not cause significant systemic effect at this dose. Previously, it was reported that 0.5 μg of sortase-ligated toxin resulted in discernable paralysis in the DAS assay, although intraperitoneal injection of 1 μg did not cause any observable effect[3]. The discrepancy in the DAS assay might be due to some inherent instability of the recombinant toxin compared to the synthetic ligated protein, or it is possible that the considerable amount of free LH$_N$/X leftover in the ligation sample contributed to the original observed muscle paralysis. In order to verify this hypothesis, the toxicity of the LH$_N$/X fragment was evaluated in another DAS assay ($n$ = 3, Fig. 1e and Supplementary Table 4). The mice developed partial paralysis after 24 h, demonstrated by a reduced ability to spread toes following a startle stimulus. LH$_N$/X that had not been activated with trypsin served as a control and did not show any effect at similar doses. These results support that the higher activity previously reported with the sortase-ligated BoNT/X material was likely due to the large amounts of non-ligated, activated LH$_N$/X in the sample[3].

### Cryo-EM structure of BoNT/X in complex with NTNH (M-PTC/X)

We have determined the cryo-EM structure of the M-PTC/X complex comprising the BoNT/X R360A/Y363F inactive mutant holotoxin and its non-toxic interaction partner NTNH/X at a nominal resolution 3.1 Å (using the gold-standard FSC criterion of 0.143), obtained from 432,063 particles selected from a total of 5408 recorded movies from two datasets in similar data acquisition settings, described in the "Materials and methods" section (see Supplementary Fig. S3 and Supplementary Table 5). The overall

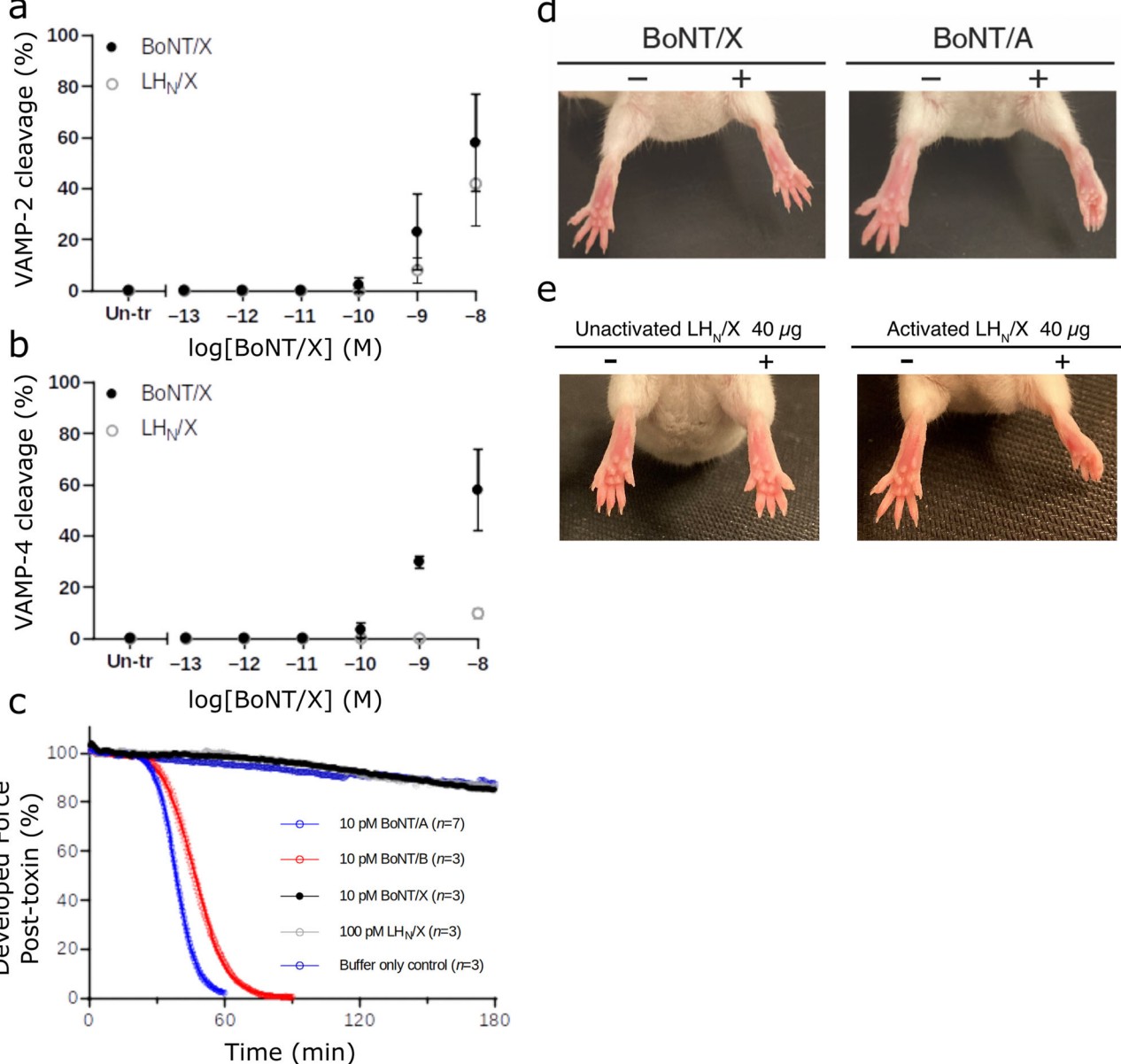

**Fig. 1 | Activity of BoNT/X.** Rat cortical neurons were exposed to varying concentrations of recombinant full-length BoNT/X (filled circles) or $LH_N X$ (open circles). After 24 h of incubation cells were lysed, and lysates analyzed for VAMP2 (**a**) and VAMP4 (**b**) by Western blots (error bars represent standard deviation from $n = 3$ replicates). **c** Mouse phrenic nerve hemidiaphragm (mPNHD) preparations were exposed to 10 pM BoNT/X, BoNT/A, and BoNT/B, 100 pM $LH_N/X$, and muscle contraction responses to indirect stimulation recorded until contraction was no longer detectable or to 180 min ($n = 3$). **d** BoNT/A (6 pg) induced flaccid paralysis when injected into the right gastrocnemius muscle of mice in DAS assays, whereas injection of BoNT/X (1 μg) did not induce muscle paralysis. **e** $LH_N/X$ (40 μg) was injected to the right gastrocnemius muscle of mice ($n = 3$). The limb developed partial paralysis after 24 h (DAS Score = 2 out of 4) and the toes showed a reduced ability to spread following a startle stimulus. $LH_N/X$ that had not been activated with trypsin served as a control. A representative image of paralysis obtained from injection of 40 μg is shown and experiment data are shown in Supplementary Table 4.

structure of the BoNT/X-NTNH/X complex is shown in Fig. 2a–c which shows an assessment of the local resolution throughout the entire map, demonstrating that the core of the complex has a resolution better than 3.1 Å. Figure 2d–f provides visual examples of the map quality.

BoNT/X and NTNH/X share a similar fold and their overall domain arrangement and molecular architecture is similar to other toxin serotypes (Fig. 2a, b and Supplementary Fig. S4). Similarly to the known M-PTC structures of the A and E serotypes, BoNT/X and NTNH/X form an interlocked complex that buries a large solvent-accessible area of 4778 Å², which is significantly larger compared to BoNT/A-NTNH/A (3664 Å²) and BoNT/E-NTNH/E (3459 Å²). The main domains that are involved in this interaction are the receptor-binding domain in BoNT/X ($H_C/X$) and

the analogous domain in NTNH/X ($nH_C/X$), together with the translocation domain and its analog ($H_N/X$ and $nH_N/X$). LC and nLC are situated in both extremities of the complex, facing opposite directions from the interface, and are mostly solvent-exposed. The critical disulfide bond between C423 and C467 that holds the light and heavy chains of the toxin together after nicking by host or bacterial proteases is discernible in the map, in the same position as in the structures of BoNT/A, B, and E[13,14,24]. It has been shown that C461 can also form a disulfide bond with C423, with the toxin retaining activity in C467S mutants[3]. Our cryo-EM map indicates that C423-C467 is the most likely natural disulfide bridge. C461 is not visible in the map, as it is located in a loop that is either nicked or disordered. In our structural studies, we used a catalytically inactive double

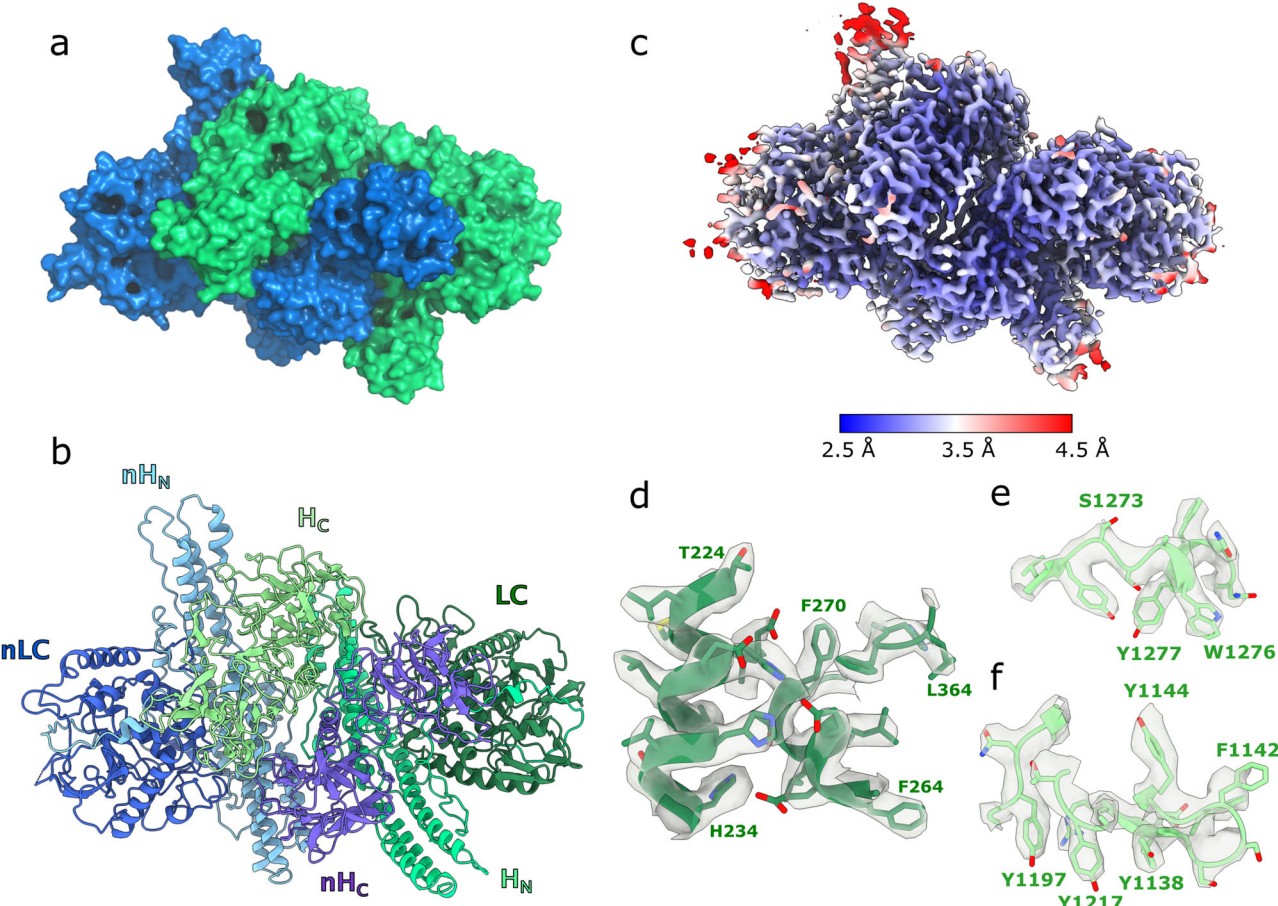

**Fig. 2 | Cryo-EM structure of BoNT/X in complex with NTNH/X. a** Surface view of the BoNT/X (green)—NTNH (blue) complex. **b** Cartoon view of the different domains in each protein. BoNT/X domains are colored in green shades, NTNH/X domains are colored in blue shades. **c** Local resolution estimates (Å) for the cryo-EM map of the M-PTC/X. **d** cryo-EM map around the LC active site. **e** cryo-EM map around the SxWY ganglioside-binding motif. **f** cryo-EM map around the $H_C$ patch rich in aromatic residues.

mutant construct of BoNT/X described in the "Materials and methods" section. The mutated residues are not involved in the coordination of the $Zn^{2+}$ atom, and the map in the area between residues E226, H227, and H231 indicates that the metal likely is present in the active site. At this resolution we decided not to include this ion in the model since we previously determined the 1.35 Å crystal structure of the catalytic domain of BoNT/X, describing the metal coordination in details[9].

The translocation domain consists of an antiparallel α-helix bundle with two central helices of approximately 105 Å flanked by smaller helices and linked with LC through a region known as the "belt". The belt of BoNT/X wraps around LC (Fig. 3a), likely acting as a molecular chaperone, as has been shown for BoNT/A[25]. It occupies part of a groove on the LC surface which is also associated with substrate-binding[26]. The belt adapts remarkably to the LC surface, as its presence in the full-length toxin does not affect the local structure of LC with interacting regions presenting similar conformations to those seen in the free domain structure (Fig. 3c). As in other BoNTs, the extended interaction site of the belt region with LC/X is likely to also mimic the binding of LC to its substrate proteins (VAMP1,2,3,4,5 and Ykt6) which are expected to involve multiple exosites away from the catalytic center (Fig. 3a)[9,26]. Fig. 3b shows the overlap between a VAMP-like inhibitor bound to LC/F and the $H_N$/X belt region[27].

The translocation domain shares an extensive interaction with LC beyond the belt, and presence of the central $H_N$ has a significant impact on the LC structure when compared to the free domain, as shown in Fig. 3d. On one side, the $H_N$ C-terminal linker (875–878) interacts with LC to slightly

rearrange loop 272–278 via hydrophobic interactions between P877/F878 and I273. Further, what appears as an extended loop (195–215) stabilized by two β-strands in the free LC, is seen with a much more compact conformation in the complex, with loss of the secondary structure as the I207-V208 residues are pushed by hydrophobic interaction with Y895 and Y951 on the binding domain ($H_{CN}$). These interactions between LC and $H_{CN}$ are interesting as they might also help to stabilize the binding domain in a similar position in the free holotoxin. Furthermore, a significant structural change between the free LC/X structure and the BoNT/X-NTNH/X complex was observed in the vicinity of the catalytic site where loop 243–263 takes on a long and flexible conformation extending away from the catalytic pocket so that T255 interacts with $H_N$. In the free LC/X structure, residues 243–263 take on a more complex psi-loop motif in close proximity to the catalytic pocket entrance. The plasticity of this loop may be important for substrate recognition.

**Crystal structure of free NTNH/X**

In contrast to free BoNT/X, NTNH/X exhibited high thermal stability in the toxin-free state, demonstrated by nanoDSF curves with late unfolding onsets and single, sharp inflection points over a broad pH range with the highest melting temperature of 49.5 °C at pH 6.5 (Supplementary Table 6, Supplementary Fig. S5). NTNH/X was also stable at high protein concentration (over 40 mg/ml). We determined its structure by X-ray crystallography at the resolution of 3.3 Å (Supplementary Table 7). Although the electron density map was not well defined for many side chains, we were able to build the complete

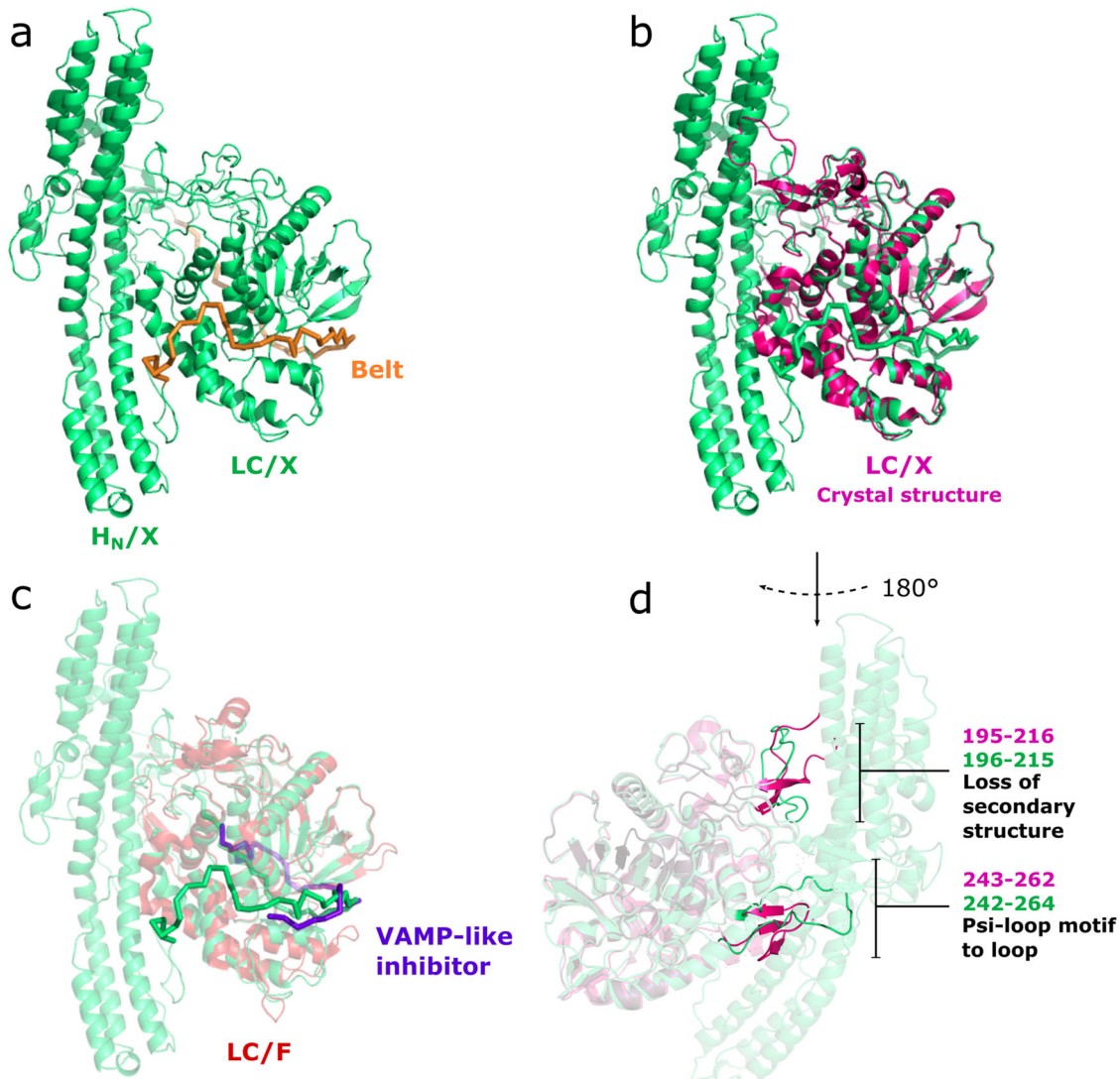

**Fig. 3 | Structure of the BoNT/X light chain. a** $H_N$ and LC domains from the BoNT/X-NTNH/X cryo-EM structure (green). The belt region of $H_N$ surrounding the LC is highlighted in orange. **b** crystal structure of LC/F (PDB ID 3FIE, colored in red) bound to a VAMP substrate-like inhibitor (purple) superimposed to $H_N$-LC/X (green). **c, d** crystal structure of LC/X (PDB ID 6F47, colored in magenta) superimposed to the cryo-EM structure of $H_N$-LC/X (green). Noteworthy differences in secondary structure are highlighted in (**d**).

main chain, including regions missing in the cryo-EM model. The overall structure of NTNH/X in the free state is very similar to that in the BoNT/X-NTNH/X complex, exhibiting the same domain organization with only a slight difference in the angle between the nLC-nH$_N$ portion and the nH$_C$ domain (Supplementary Fig. S6a). The nH$_C$ domain is mainly responsible for the binding to BoNT/X and has been shown to be flexible in solution in the X-ray and SAXS studies on NTNH/D[28]. The structures of the individual subdomains of NTNH/X only differ between the free and toxin-bound states locally, in certain loop regions involved either in the toxin-NTNH interface or in crystal contacts (Supplementary Fig. S6b–d). The high stability, relative rigidity, and domain arrangement of NTNH/X support its function as a structural scaffold for BoNT/X.

NTNH/X lacks the nLoop conserved in the nLC of NTNH/A1, B, C, D, and G, which facilitates the assembly of the L-PTC with the HA proteins (Supplementary Fig. S7)[29–31]. This correlates with the absence of HA proteins in the BoNT/X gene cluster, which is of the *orfX*-type[3]. However, NTNH/X has an extended loop with exposed hydrophobic residues, structurally close to the NTNHA nLoop. This loop (NTNH/X residues 128–138) could not be built in the cryo-EM model of the complex, but was built in the X-ray model

of free NTNH/X where it extends into the solvent (Supplementary Fig. S6b). Although not conserved in other *orfX*-type NTNH proteins, its position suggests a potential interaction site with other proteins.

### Structural differences between isolated BoNT/X and the BoNT/X-NTNH/X complex

To probe possible conformational changes in BoNT/X, we performed atomistic molecular dynamics (MD) simulations on the isolated BoNT/X as well as for the intact BoNT/X-NTNH/X complex. Our simulations reveal significant conformational changes in the H$_C$ of the isolated BoNT/X relative to the MD simulations of the BoNT/X-NTNH/X complex, which remains in a conformation similar to that observed in the experimental cryo-EM structure (see Fig. S8 and Supplementary Movies 1–2). In this regard, the isolated BoNT/X undergoes a translational motion of the helix (Glu880 to Tyr895) within the linker region connecting H$_C$ with the rest of the protein. This moves H$_C$ towards H$_N$ (Fig. S8) and leads to an overall more compact conformation relative to the BoNT/X-NTNH/X form. Our MD results align well with AlphaFold predictions of the isolated BoNT/X, where we observe a twist in the helix within the linker region, with statistical prediction scores that can be related to protein dynamics, further supporting the hinge

motion[32]. Taken together, we suggest that the isolated BoNT/X could adopt a more compact conformation compared to its complex form, different from the conformations of isolated BoNT/A and BoNT/E (Supplementary Fig. S8).

### pH-dependent stability of BoNT/X

Despite their high similarity and common domain architecture, BoNT/X exhibited a very different behavior than NTNH/X in the nanoDSF stability assay (Supplementary Table 6). Analysis of the shape of the melting curves and the Gibbs free energy of unfolding ($\Delta G_u$), which positively correlates with protein stability (Supplementary Fig. S5), confirmed that BoNT/X was significantly less stable than NTNH/X[33]. NTNH/X was more stable at acidic pH, which supports its role in protecting the toxin in such environment. In contrast, BoNT/X's instability was less pronounced at neutral to basic pH, at which it may naturally exist in the free form and perform its function. The shape of BoNT/X melting curves indicated a gradual and stepwise denaturation profile with an early onset of unfolding, particularly at low pH. At endosomal pH, BoNT/X appears to exist in a semi-molten state and to unfold domain by domain. Following receptor-mediated endocytosis, the toxin is expected to undergo extensive conformational changes upon acidification, likely accompanied by partial toxin unfolding, and ultimately resulting in translocation of the LC into the cytoplasm[34]. It would be intriguing to study the effect of pH on the interaction of BoNT/X with a lipid bilayer, especially with respect to the pH-driven events in the endosome and the translocation. However, the instability of BoNT/X and its tendency to aggregate, likely driven also by the hydrophobic patches in the receptor-binding domain, prevented us from successfully performing further structural studies.

### Analysis of the complex's pH-dependent stability

In order to release the neurotoxin upon leaving the gastrointestinal tract, the M-PTC needs to disassemble in a pH-dependent manner. The large interface in M-PTC/X contains several interaction patches that are displayed in Fig. 4a. The interface patches in BoNT/X (theoretical isoelectric point of 5.6) are predominantly positively charged and those in NTNH/X (theoretical isoelectric point of 4.6) are mostly negatively charged. The M-PTC/X is therefore stabilized by a number of electrostatic interactions between BoNT/X and NTNH/X shown in Fig. 4b. A cluster of solvent-accessible acidic and basic residues can be found on the interface between $H_{CN}/X$ and $nH_N/X$, namely D998, K1000, K1046, E1048, E1049, K1050, D1051, K1013, and D1018 in BoNT/X; and K636, E787, E789, and E793 in NTNH/X. The location of this pH-sensing cluster corresponds to a similar cluster in BoNT/A-NTNH/A, which has been demonstrated to drive the pH-mediated disassembly of the M-PTC/A[13]. The charge of these residues will depend on the environmental pH, which was pH 5.5 in the buffer utilized for the cryo-EM experiments. It is not trivial to predict the pH at which the protonation state of the residues in this pH-sensing cluster will change, since the $pK_a$ of the amino acids strongly depends on their local chemical environment[35]. However, protonation states of titratable residues were predicted based on an electrostatic model using PropKa3.0 performed for the BoNT/X-NTNH/X complex and the individual BoNT/X and NTNH/X subunits (Supplementary Table 9). All residues were predicted to be in their standard protonation states for the isolated BoNT/X, whilst in the BoNT/X-NTNH/X complex BoNT/X Glu1048 and NTNH/X nGlu554, nGlu584 were predicted to be in their protonated form.

In order to determine the pH at which the complex disassembles, we analyzed size-exclusion chromatography profiles of BoNT/X-NTNH/X at pH 5.5, 6.5, 7.5, 8.5, and 9.5. The chromatograms are shown in Fig. 4c. The two peaks in the chromatograms correspond to the BoNT/X-NTNH/X complex and to a mixture of free BoNT/X and free NTNH/X, respectively. The complex is stable at pH 5.5, 6.5 and 7.5. At pH 8.5, the equilibrium is shifted towards free BoNT/X and NTNH/X, and at pH 9.5 the complex is almost totally disassembled. To further evaluate the stability and function of the BoNT/X-NTNH/X complex, we then carried

out limited proteolysis studies with trypsin treatment at a range of pH conditions (pH 5.0, 6.0, 7.5, and 8.0). As shown in Fig. 4e, f, we found that trypsin easily degraded BoNT/X and NTNH/X at all pH conditions, whereas the BoNT/X-NTNH/X complex is largely resistant to trypsin at pH 5.0, 6.0, and 7.5, demonstrating that forming the complex protects BoNT/X from proteases at these pH conditions. After trypsin treatment, BoNT/X within the complex is separated into two fragments of ~100 and ~50 kDa on SDS-PAGE gels (Fig. 4e, f): this is likely the HC and LC of BoNT/X as trypsin can still cleave the linker region between the HC and LC[36]. This protection from trypsin was not observed at pH 8.0 for the complex, suggesting that the complex is not stable at basic conditions, which is consistent with our analysis by size-exclusion chromatography.

Dissociation of the BoNT/X-NTNH/X complex occurs at a significantly more basic pH than BoNT/A-NTNH/A, which is stable at pH 6.0 but disassembled at pH 7.5[13]. These results, together with its relatively low toxicity in mice, make it tempting to speculate that BoNT/X might target an organism with a different gut environment. Insects for instance have a more alkaline digestive system, and anopheles mosquitoes are known to be the target of the BoNT-like PMP1[11]. However, BoNT/X was discovered in a clostridial strain associated to a clinical case of human botulism.

### Characterization of the receptor-binding domain of BoNT/X

Although BoNT/X was shown to induce flaccid paralysis in mice at high concentrations, its neuronal receptors remain elusive[3]. The M-PTC/X structure presented in this study provides the first insights of the toxin's receptor-binding domain ($H_C/X$). Overall, $H_C/X$ presents the same architecture observed across all clostridial neurotoxins, which include the lectin-like fold of the N-terminal subdomain ($H_{CN}/X$), which is essential for toxicity[37], and the C-terminal subdomain ($H_{CC}/X$). This β-trefoil fold is the main element responsible for neuronal recognition.

The binding domain position has been observed to vary in the different holotoxins, from a linear configuration of the three functional domains in BoNT/A and B to a more compact, closed, structure in BoNT/E where the $H_C$ interacts with both the $H_N$ and the LC[38]. This unique position of $H_C/E$ was suggested to promote the faster translocation rate of BoNT/E[39]. Remarkably, the position of the $H_C$ within the M-PTC is consistent across all serotypes, where it is stabilized by its interaction with NTNH into an intermediate configuration (Supplementary Fig. S4)[13,14]. Mobility of the binding domain is supported by a flexible linker between $H_N$ and $H_C$, which involves a short helical component (Fig. 4d). Although there is no particular sequence conservation for that region, BoNT/X in the M-PTC structure also presents a flexible linker followed by a small helical conformation leading to $H_C$ (residues 876–895). Interestingly, this segment is located between the two LC and $H_C$ interaction sites that were mentioned previously, and that mainly involve hydrophobic interactions (Fig. 4d).

Canonical BoNTs (BoNT/A-BoNT/G) and the tetanus neurotoxin all have gangliosides as receptors or co-receptors. The toxins recognize these cell surface carbohydrates through a shallow pocket centered around the conserved SxWY ganglioside binding motif on $H_{CC}$[40], which is also present in the primary sequence of BoNT/X[3]. Superposition of $H_C/X$ from our structure with ganglioside-bound $H_C/A$ (PDB ID 5TPC, rmsd of 1.711 Å over 388 Cα pairs, Fig. 5a) reveals that the SxWY motif is oriented similarly to that in $H_C/A$, suggesting it could accommodate a carbohydrate moiety. However, the variation in ganglioside specificity and affinity observed in BoNTs is mainly driven by non-conserved residues surrounding the common binding motif[41], making it difficult to predict a preferred ganglioside receptor for BoNT/X.

Binding of BoNT/X to gangliosides GT1b, GD1b, GD1a, and GM1 was analyzed (Fig. 5d-g) using a plate-based assay as described previously[42,43]. BoNT/X was used rather than $H_C/X$ because of the reported instability of this domain when expressed individually. Despite presenting the consensus SxWY motif, BoNT/X showed very weak binding to all assayed molecules, which are the most abundant mammalian neuronal gangliosides[44]. As expected, the BoNT/A1 control presented strong binding to GT1b and GD1a, but not to GD1b and GM1[45]. Interestingly BoNT/X presented a consistently higher affinity than

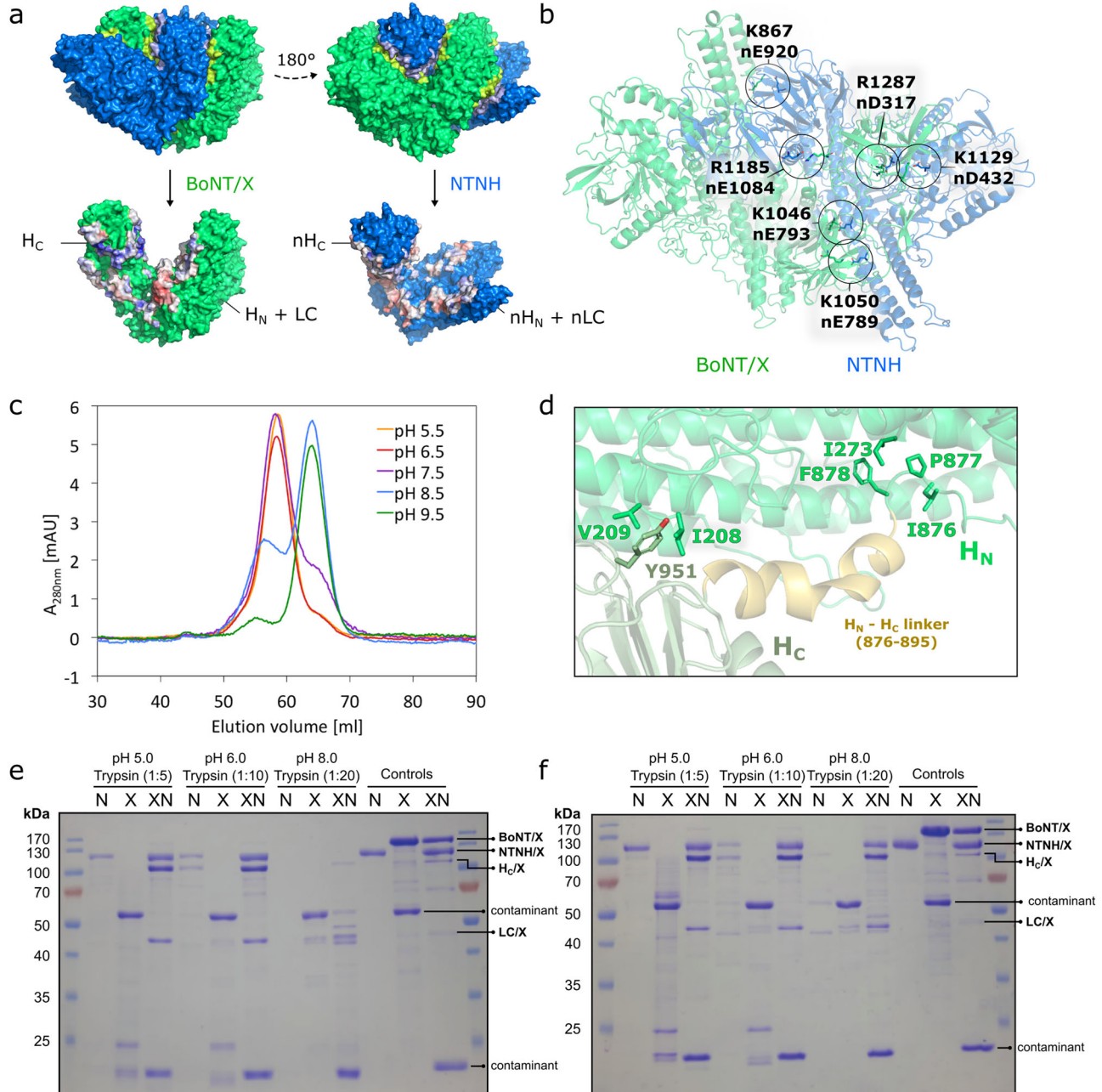

**Fig. 4 | BoNT/X-NTNH/X complex stability. a** BoNT/X-NTNH/X interface; BoNT/X is shown in green and NTNH/X in blue. Top: interfacing residues are highlighted in different shades of green and blue. Bottom: residues at the interface of the complex are colored according to electrostatic potential (negative electrostatic potential is colored in shades of red, and positive in shades of blue). **b** Electrostatic interactions across the BoNT/X-NTNH/X interface. **c** Size-exclusion chromatograms from the M-PTC/X pH stability experiments. The $A_{280nm}$ response for the chromatograms at pH 5.5 and 6.5 was divided by the factor of two to normalize it to the other chromatograms, where the signal was lower. **d** Flexible linker between $H_N$ and $H_C$ followed by a small helical region leading to the $H_C$ (residues 876–895). **e, f** Limited proteolysis assay for examining the stability of the BoNT/X-NTNH/X complex at different pH conditions. BoNT/X (marked as X), NTNH/X (marked as N), and BoNT/X-NTNH/X complex (marked as XN) were incubated with trypsin at the indicated pH conditions and then analyzed by SDS-PAGE and visualized by Coomassie blue staining. BoNT/X alone and NTNH/X alone were degraded at all pH conditions. BoNT/X-NTNH/X is largely resistant to trypsin at pH 5.0, 6.0, and 7.5, but not at pH 8.0. Trypsin treatment is able to cleave the linker between LC and HC of BoNT/X within the BoNT/X-NTNH/X complex, thus BoNT/X showed as two separate bands (LC/X and HC/X) after the disulfide bond connecting the LC and HC is reduced. One contaminant is present in purified BoNT/X and another is present in purified BoNT/X-NTNH/X complex.

BoNT/A1 for gangliosides that lack the Sia5 moiety (GD1b and GM1) (Fig. 5). Although these results are not entirely conclusive, they hint at a natural carbohydrate-binding capacity for BoNT/X. Ganglioside recognition is considered an essential first step in BoNT toxicity as it provides abundant anchorage on neuronal cell surface where the toxin can then attach to their high-affinity protein receptors for uptake[40]. The weak binding of BoNT/X might help explain the low toxicity observed so far in mammals.

BoNT serotypes A[46–48], D[49], E[50], and potentially F[42,51] are known to bind to synaptic vesicle glycoprotein 2 (SV2). Structural details on the interaction of the toxin type A with SV2 are available through the crystal structures of its binding domain ($H_C$/A) in complex with the luminal loop 4 of SV2C[48,52–54]. The $H_C$/X backbone superposes well overall with the glycosylated SV2C-bound $H_C$/A (PDB ID 5JLV), presenting an rmsd of 1.915 Å over 388 Cα pairs, differing mostly in the $H_{CN}$ region. In $H_C$/A, SV2 recognition occurs through a complementary beta-sheet between the SV2C and a beta-hairpin

**Fig. 5 | Investigation of potential ganglioside binding by BoNT/X. a** Alignment of $H_C$/X (green) and GD1a-bound $H_C$/A (in grey, PDB ID 5TPC, GD1a shown in grey sticks), showing the conserved ganglioside-binding SxWY motif residues and (**b**) the same alignment showing the exposed region rich in aromatic residues. Location of the ganglioside-binding motif (yellow) and the hydrophobic patch (orange) is shown in the context of $H_C$/X in (**c**). Binding of BoNT/X (green), and $H_C$/A (grey) to GT1b (**d**), GD1a (**e**), GM1 (**f**), and GD1b (**g**), error bars represent standard deviation from $n = 3$ replicates. Glycobloc schematic representation is shown for each ganglioside[90].

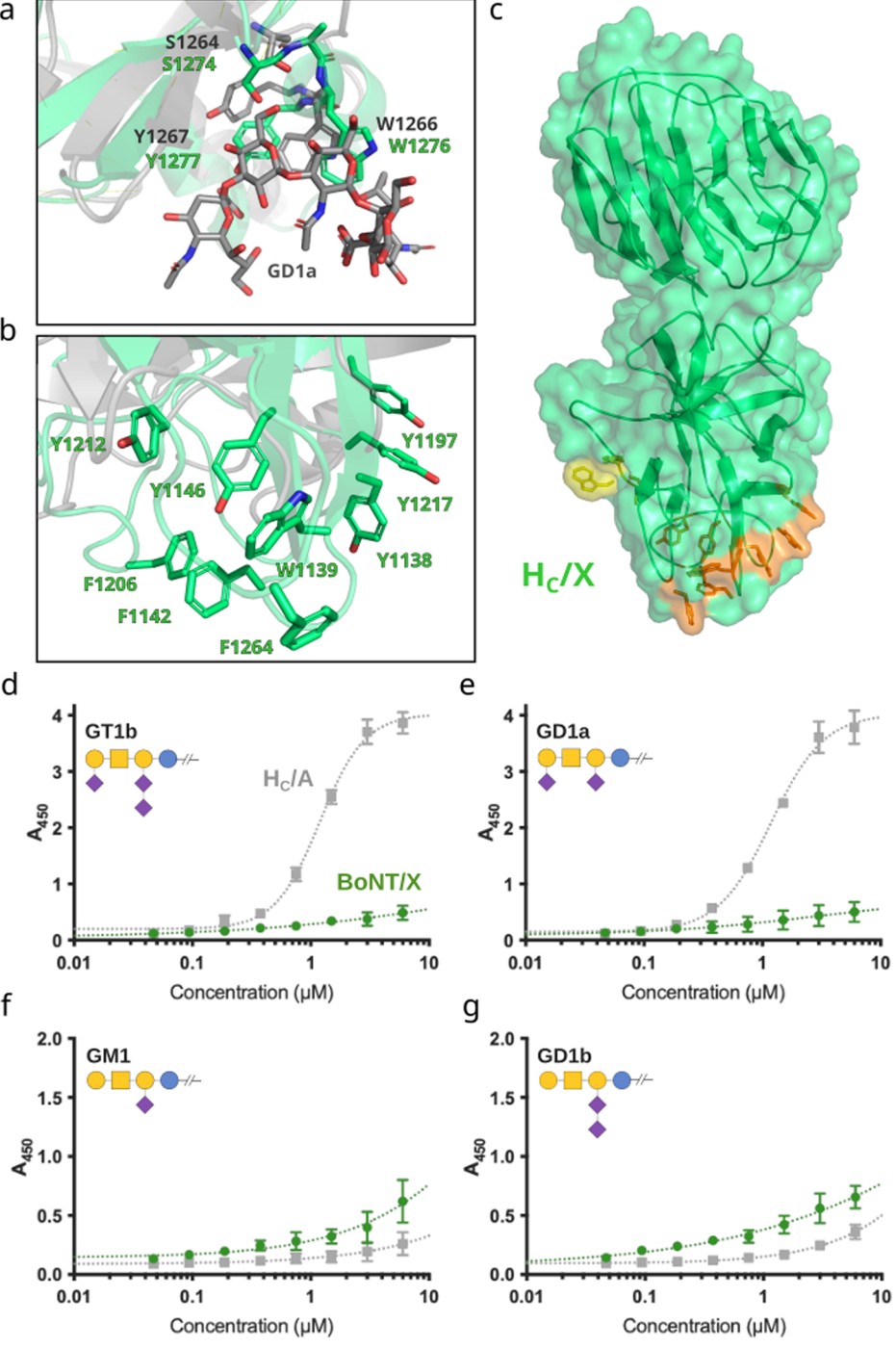

loop of $H_C$. Notably, the corresponding site in BoNT/X shows no sequence conservation and no analogous secondary structure. In addition, residues F953 and H1064 of $H_C$/A that form stacking interactions with the N559-glycan of glycosylated SV2C, which was shown to be essential for neuronal recognition, are also not conserved in $H_C$/X.

Structurally, $H_C$/X is more similar to $H_C$/D, $H_C$/E, and $H_C$/F than $H_C$/A, however, there is no obvious conservation of a potential binding site for SV2. Rmsd values for these superpositions are summarized in Supplementary Table 8.

The other principal BoNT protein receptors identified so far are synaptotagmins I and II for BoNT/B, DC, and G[55–60]. BoNT/B and DC bind to synaptotagmin with varying affinity and at two distinct sites located perpendicularly to each other[61–63]. $H_C$/X would need to undergo significant conformational changes in order to accommodate synaptotagmin at the

equivalent sites, since loops 1139–1144 overlap with the $H_C$/B synaptotagmin binding site, and loops 1201–1209 and 1247–1252 occupy the $H_C$/DC synaptotagmin binding site. The structures of free and synaptotagmin-bound $H_C$/B only show minor structural differences (agreeing with an rmsd of 0.566 Å over 441 Cα pairs, PDB IDs 2NM1 and 1Z0H)[61,62,64], suggesting that a significant conformational change upon receptor-binding is unlikely in $H_C$/X.

Remarkably, $H_C$/X has a pronounced patch of exposed aromatic and hydrophobic residues, which is thermodynamically very unfavorable and uncommon in soluble proteins unless it has an important function. This hydrophobic patch on the $H_C$/X loops, including residues Y1138, W1139, F1142, Y1146, Y1197, F1206, Y1212, Y1217, and F1264 (shown in Fig. 5b,c), may be involved in neuronal recognition via interactions with a protein or glycosphingolipid receptor, or with the phospholipid-bilayer

of the neuronal membrane itself. This hydrophobic patch in BoNT/X is located at the same position in the binding domain as where BoNT/B binds its protein receptor[61,62,65]. Interestingly, PMP1 has a pronounced hydrophobic patch with similar properties, located at a similar position in the binding domain as BoNT/X, even though the overall sequence conservation is low[11].

Hydrophobic interactions directly with the membrane have recently been described as complementing the receptor-binding capacity of BoNT/B, G, and DC[1,45,61,62,66,67]. However, the hydrophobic patch on the binding domains of PMP1 and BoNT/X are structurally distinct from the single, extended, hydrophobic loop that contributes to anchoring these toxin serotypes to the membrane.

## Concluding remarks

In this study we solved the cryo-EM structure of the 300kDa BoNT/X-NTNH/X assembly, showing that cryo-EM is a highly suitable method to analyze M-PTC complexes, and facilitate studies on full-length toxins that are problematic to characterize by themselves. We provide a template for the structural analysis of these complexes using single particle cryo-EM and demonstrate its usefulness for future studies of other BoNT-NTNH systems. We were able to reach similar resolution to the X-ray crystallography structures determined for BoNT/A-NTNH/A and BoNT/E-NTNH/E, without the significant challenge of obtaining crystals of sufficient quality or the use of nanobodies to facilitate crystallization[13,14]. The BoNT/X-NTNH/X complex structure also provides structural information on the receptor-binding domain of BoNT/X, which revealed an unusual array of surface-accessible hydrophobic patches despite presenting the same overall fold as other BoNTs. Although $H_C/X$ presents a conserved carbohydrate-binding motif, it did not bind significantly to any of the most abundant human gangliosides. The structure hence hints at a unique mechanism of cell recognition.

Stability of the M-PTC is known to be pH-dependent to promote oral toxicity and release of the toxin once it has crossed the intestinal barrier. Assessment by size-exclusion chromatography and protease assays showed that the complex remains stable up to pH 7.5, which is an unusual property contrasting with BoNT/A-NTNH/A that dissociates at a lower pH. The structure revealed patches of acidic and basic residues at the interface between $H_C/X$ and $nH_N/X$ that are likely to substantially contribute to the pH-sensing mechanism of the complex disassembly. NTNH certainly provides stability for BoNT/X and thus the M-PTC could be a valuable component in the production and formulations of BoNT/X-derived biologicals.

BoNT/X presented very weak activity on cultured neurons and in vivo, even more so than in preliminary reports[3]. The native full-length BoNT/X used in this study provides a more accurate assessment of its toxicity, as we show that the $LH_N/X$ fragment causes similar levels of paralysis as those previously reported for sortase-ligated BoNT/X, which contained very large amounts of non-ligated $LH_N/X$. During the revisions of this manuscript, a study supporting the very low in vivo potency of BoNT/X was published.[91] This study also showed that strain 111 produces full-length, catalytically active, BoNT/X.[91] The LC of BoNT/X was shown to have a particularly high catalytic activity[9], and here the $LH_N/X$ fragment was shown to retain a functional level of toxicity on neuronal cells which was comparable to the same fragment of other serotypes[19,21]. In addition, exchanging the binding domain of BoNT/X for $H_C/A$ produced a functional and efficient toxin[8]. Altogether, the low toxicity of BoNT/X is likely linked to the unusual properties of its binding domain, including the hydrophobic surface and unclear receptor-recognition strategy, particularly towards gangliosides. This suggests an absence of suitable BoNT/X receptors on motor neurons of mammalian species.

BoNT/X was identified from a clinical isolate but is evolutionarily closer to other recently identified botulinum-like toxins, targeting insects (PMP1), or isolated from bovine feces (BoNT/En). It would therefore not be surprising if the true target of BoNT/X are non-mammalian species or non-neuronal cells. However, the medical and biotechnological potential of the functional $LH_N$ fragment of BoNT/X is evidently vast and warrants further studies[7,8,68].

## Materials and methods
### Purification of full-length, active BoNT/X
pK8-BoNT/X(Cloop)-10HT encodes BoNT/X full-length (UniprotKB P0DPK1) with the activation loop (C423-C467) substituted with the activation loop of BoNT/C (Uniprot KB P18640, C437—C453) and a C-terminus PreScission-cleavable H10 tag in the pK8 expression plasmid (pET26b derived synthesized at Entelechon). cDNAs were codon-optimized for E. coli expression and synthesized by GeneArt in two fragments for biosafety reasons: the LC of BoNT/X (M1—I422) with the activation loop of BoNT/C cloned into pK8 and the heavy chain of BoNT/X (I468—D1306) with a C-terminal PreScission-cleavable H10 tag. The heavy chain was then cloned in pK8 in the C-terminus of the 1st fragment using BsaI for scarless insertion.

NTNH/X (NCBI RefSeq WP_045538950) was cloned from pET28a-NTNH/X into pET32a using NdeI and XhoI restriction enzymes.

pET32a-NTNH/X and pK8-BoNT/X(Cloop)-10HT were co-transformed into NiCo21(DE3) competent E. coli (NEB). Protein expression was induced with 1 mM IPTG at 16 °C for 20 h in mTB (Casein Digest Peptone 12 g/l, Yeast Extract 24 g/l, Dipotassium Phosphate 9.4 g/l, Monopotassium Phosphate 2.2 g/l, 0.4% Glycerol) supplemented with Kanamycin and Ampicillin. Cells were harvested at 4 °C and lysed by sonication in 50 mM HEPES pH 5.5, 300 mM NaCl, 10 mM Imidazole, treated with Benzonase nuclease (Sigma), and clarified by centrifugation. Proteins were purified using the ÄKTA Pure FPLC system (Cytiva). Cleared lysates were loaded into HisTrap HP columns (Cytiva). The protein complex was eluted with 50 mM HEPES pH 5.5, 300 mM NaCl, 500 mM Imidazole and separated by desalting into a higher pH buffer (50 mM HEPES pH 7.5, 300 mM NaCl) using a HiPrep Sephadex G-25 desalting column (Cytiva). After concentration to 1 mg/ml, activation was performed overnight at 4 °C with 5 μg of Factor Xa (NEB) per mg of protein. To separate activated BoNT/X(Cloop)-10HT from NTNH/X, the activated sample was loaded onto a HisTrap HP column equilibrated in 50 mM HEPES pH 7.5, 300 mM NaCl, and eluted in two steps at 500 and 1000 mM Imidazole. NTNH/X was found in the flow through and BoNT/X(Cloop)-10HT in elution fractions, which were pooled and buffer exchanged and adjusted to 0.1 mg/ml into PBS (Gibco) supplemented with 1 mg/ml BSA.

### Measurement of VAMP cleavage in cortical neurons
CTX cells were treated with serial dilutions of BoNT and incubated at 37 °C for 24 h, following 3 weeks in vitro. Cells were lysed by removing all medium and adding sample buffer (25% NuPAGE buffer (Life Technologies, Fisher Scientific) supplemented with 10 mM dithiothreitol and 250 units/μl Benzonase (Sigma)). Lysate proteins were separated by SDS-PAGE and transferred to nitrocellulose membranes. Primary antibodies used were against VAMP-2 (custom-made) and VAMP4 (Santa Cruz sc-365332). The secondary antibodies were HRP-conjugated ant-rabbit IgG (Sigma A6154) and HRP-conjugated anti-mouse IgG (Sigma A4416). Proteins were visualized using an enhanced chemiluminescent detection system (Fisher Scientific). Luminescence detection was carried out using a Syngene GeneGnome and image analysis was performed using Gene-Tools software (Syngene Bioimaging, Cambridge UK). VAMP cleavage was monitored by measuring the disappearance of the specific full-length VAMP protein and the appearance of the cleaved fragment of the VAMP protein. The amount of cleaved VAMP protein was expressed as a percentage of the sum of full-length protein and cleaved product when available.

### Mouse phrenic nerve hemidiaphragm assay
The mPNHD assay is an isolated, ex vivo model used to measure the effect of botulinum neurotoxin (BoNT) at its in vivo target, the NMJ. Male CD1 mice, 25–30 g (Charles River Laboratories, UK), were killed by $CO_2$ asphyxiation, and the hemidiaphragm muscle and attached phrenic nerve

isolated and attached to a custom tissue holder/electrode (Emka Technologies, France) installed in a 10 ml tissue bath (EmkaBATH4 Tissue Bath System, Emka Technologies, France) containing Krebs–Henseleit buffer (KHB; 118 mM NaCl, 1.2 mM MgSO$_4$, 11 mM Glucose, 4.7 mM KCl, 1.2 mM KH$_2$PO$_4$, 2.5 mM CaCl$_2$, 25 mM NaHCO$_3$, pH 7.5 (Sigma, UK) at 37 °C) gassed with Carbogen (95% O$_2$/5% CO$_2$; BOC, UK). The phrenic nerve was continuously electro-stimulated using 10 V, 1 Hz, 0.2 ms stimulation, and resultant muscle contractions, as a result of Acetylcholine (ACh) release at the NMJ, were recorded using an isometric force transducer (Emka Technologies). The preparations were allowed to equilibrate for 45 min in KHB renewed every 15 min. Following equilibration, the tissue was incubated with 3 μM tubocurarine hydrochloride (Sigma), a reversible, competitive antagonist at the nicotinic ACh receptor on the muscle cells. Subsequent inhibition of muscle contraction was considered an indication that the contraction response was due to ACh released by nerve stimulation (data from preparations with inhibition below 95% were discounted). The preparations were then washed 3× for 5 min and after a further stabilization period of 20 min, 1 ml of 10x BoNT (final concentration 10 pM for BoNT/A and BoNT/B (both LIST Biological Laboratories, Campbell, USA), and recombinant (r)BoNT/X, and 100 pM for LH$_N$/X, in KHB supplemented with 0.05% (w/v) gelatin type A (Sigma) was added to the tissue bath and electrical stimulation continued until muscle contraction was ablated. Following complete paralysis, the muscle was directly stimulated (bypassing the phrenic nerve) by increasing the strength of the applied electro-stimulation (10 V, 1 Hz, 2 ms) in order to confirm the continued physiological viability of the preparation. Experimental data were recorded with IOX software (Emka Technologies). The decrease in contraction with time following toxin addition was calculated as a percentage of the contraction just before toxin addition and a four-parameter logistic curve fitted to the data, where appropriate, using GraphPad Prism (GraphPad Prism version 6.07 for Windows, GraphPad Software, La Jolla, CA, USA, www.graphpad.com). From the curve fitted to the data, the time to 50% diaphragm paralysis (t50) was estimated.

### Digital abduction score (DAS) assay

All procedures were conducted in accordance with the guidelines approved by the Institute Animal Care and Use Committee (IACUC) at Boston Children's Hospital. Female mice were purchased from Envigo (CD-1 strain, 17–20 g). BoNTs were prepared in 0.2% gelatin-phosphate buffer (pH 6.3). Mice were anesthetized with isoflurane (3–4%) and injected with BoNT/A (6 pg) or BoNT/X (1 μg) by intramuscular injection into the gastrocnemius muscle of the right hind limb using a 30-gauge needle attached to a Hamilton syringe. Muscle paralysis and the spread of the hind paw in response to a startle stimulus were observed and recorded. The hind limb digit abduction reflex in the mouse was induced by grasping the animal lightly around the torso and lifting it swiftly into the air or by lifting it with the nose pointing downwards. Animals were prescreened for a normal digit abduction response before the experiment and those showing abnormal digit abduction responses or hind paw deformities were excluded from the study. Typically, the percentage of animals with abnormal digit abduction responses is less than 1%. The digit abduction response of each mouse was scored live using a five-point scale, from normal reflex/no inhibition (DAS 0) to full inhibition of the reflex (DAS 4)[23]. Mice were scored for digit abduction response 8 h following BoNT injection, twice a day during the first-week post-dosing, and once daily for 25 days, following injection of BoNT/A. We also measured body weight daily immediately after the assessment of the digit abduction. For the analysis of LH$_N$/X, LH$_N$/X was activated by trypsin (20:1 molar ratio of LH$_N$/X to trypsin) cleavage for 30 min at 37 °C, the degree of activation was confirmed by Coomassie staining of an SDS-PAGE gel. Subsequently, 5, 10, or 40 μg of the activated toxin was injected into the mouse gastrocnemius. The mice were observed for 48 h following injection, maximal paralysis was obtained at 24 h post-injection. The toxins were compared to LH$_N$/X that had not been activated with trypsin.

### Production of BoNT/X-NTNH/X for structural studies

The BoNT/X-NTNH/X complex was recombinantly co-expressed from a pET-22b vector encoding His$_6$-tagged R360A/Y363F inactive mutant of BoNT/X in a single chain and from a pET-28a(+) vector encoding NTNH/X in a single chain (GenScript). The expression was performed in E. coli BL21(DE3) in Terrific Broth using the LEX bioreactor system (Harbinger Biotech, Toronto, Canada) and it was induced with 1 mM IPTG at OD$_{600nm}$ of 1. The cells were harvested after around 20 h of post-induction cultivation at 18 °C and lysed by sonication in lysis buffer (50 mM MES, 500 mM NaCl, 25 mM imidazole, 5% glycerol, 1 mM DTT, pH 5.5 plus complete EDTA-free protease inhibitor cocktail (Roche, Basel, Switzerland)). The BoNT-X/NTNH complex was purified by gravity-flow affinity chromatography on Ni-NTA agarose resin (Macherey–Nagel, Düren, Germany) in lysis buffer, the wash buffer contained 70 mM imidazole, and the his-tagged complex was eluted in three steps by 100 mM, 250 mM, and 500 mM imidazole in the lysis buffer. Fractions containing the BoNT/X-NTNH/X complex were pooled and dialyzed against the size-exclusion chromatography buffer (50 mM MES, 500 mM NaCl, 5% glycerol, 1 mM DTT, pH 5.5), concentrated, and further purified by size-exclusion chromatography (SEC) on a Superdex200 16/60 column and ÄKTA Pure system (GE Healthcare, Uppsala, Sweden). The pure BoNT/X-NTNH/X complex was concentrated to 8.4 mg/ml, flash frozen, and stored at −80 °C.

### SEC pH stability experiments

Fractions from the previous preparatory SEC with BoNT/X-NTNH/X were pooled, concentrated, and loaded on a Superdex 200 16/60 column (GE Healthcare, Uppsala, Sweden) a total of five times, with the column equilibrated at different pH values (pH 5.5, 6.5, 7.5, 8.5, and 9.5). Each load contained approx. 0.25 mg of the complex. The SEC running buffers are listed in Table 1.

### pH-dependent trypsin proteolysis

Recombinantly purified NTNH/X, BoNT/X, and the BoNT/X-NTNH/X toxin complex were subjected to limited proteolysis with trypsin for 13 h at room temperature. The trypsin digestions were performed at four different pHs in buffers containing 50 mM MES (pH 5.0), sodium phosphate (pH 6.0), Tris (7.5 or 8.0), and 150 mM NaCl. The trypsin:protein sample ratios (w/w) were 1:5 (pH 5.0), 1:10 (pH 5.0 or 6.0), or 1:20 (pH 7.5 or 8.0): trypsin concentrations were raised at low pH conditions as trypsin showed reduced proteolytic activity under lower pH conditions. As controls, each protein was prepared and incubated in parallel, but without trypsin. The digestion was stopped by boiling samples in reducing SDS-loading buffer for 5 min. All samples were subjected to SDS-PAGE, followed by Coomassie blue staining and destaining.

### EM sample preparation

The BoNT/X-NTNH/X sample was diluted with 50 mM MES, 150 mM NaCl pH 5.5 and blotted onto the grids using the Vitrobot blotting robot (FEI, Hillsboro, OR, USA) at 100% humidity and 22 °C, waiting for 15 s for the sample to equilibrate before blotting for 1.5 s. Grids were clipped and stored in liquid nitrogen until further analysis. For the first data set, Quantifoil R2/2 Cu 400 mesh holey carbon grids (Quantifoil Micro Tools, Jena, Germany) coated with a 0.4 mg/ml graphene oxide suspension

**Table 1 | Buffer composition for SEC pH stability experiments**

| pH | Buffer composition |
|---|---|
| 5.5 | 50 mM MES, 150 mM NaCl, 0.5 mM TCEP |
| 6.5 | 50 mM MES, 150 mM NaCl, 0.5 mM TCEP |
| 7.5 | 50 mM HEPES, 150 mM NaCl, 0.5 mM TCEP |
| 8.5 | 50 mM Tris, 150 mM NaCl, 0.5 mM TCEP |
| 9.5 | 50 mM CHES, 150 mM NaCl, 0.5 mM TCEP |

(Sigma-Aldrich, St. Louis, MO, USA) were used and the BoNT/X-NTNH/X concentration was 0.1 mg/ml. For the second data set, Quantifoil R1.2/1.3 Cu 300 mesh holey carbon grids (Quantifoil Micro Tools, Jena, Germany) without a graphene oxide coating were used and the BoNT/X-NTNH/X concentration was 0.5 mg/ml.

## EM data acquisition and processing

Two data sets were acquired on Titan Krios microscopes (FEI, Hillsboro, OR, USA) equipped with K2 summit direct electron detectors (Gatan, Pleasanton, CA, USA) and operating at 300 kV, using identical data collection parameters. The first data set was collected at the Stockholm node of the Swedish National Cryo-EM facility (SciLifeLab, Stockholm, Sweden) and the second data set at the Umeå node (Umeå University, Umeå, Sweden). Movies were acquired at 130,000× nominal magnification with a pixel size of 1.05 Å/pixel, defocus range of $-1.8$ to $-3.4$ μm in 0.2 μm steps, and a total exposure time of 10 s over 40 frames; the total dose was 35.6 and 35.9 electron/Å$^2$ for the first and second data set, respectively. A detailed explanation of the downstream processing workflow is presented in Supplementary Fig. S3. Data processing was carried out in *cryoSPARC* version 2.10[69,70]. Patch motion correction and patch CTF estimation were carried out using the respective *cryoSPARC* algorithms. In the first data set, 2885 movies were recorded from which 70 were rejected. A total of 591,151 particles were picked in cryoSPARC through iterative blob and template picking optimization, and after 2D classification 100,920 particles were selected for 3D refinement. In the second dataset, 1,123,595 particles were picked using the 2D templates from the first dataset, from 2523 recorded movies, and 351,194 particles were selected in 2D classification and used for 3D refinement. Finally, the particle sets from both datasets were combined for another iteration of 2D classification and subsequent homogeneous 3D refinement, which included 432,063 particles and yielded the final map at 3.12 Å resolution, calculated based on the gold-standard FSC of 0.143[71]. Half maps, FSC curves, and Euler angle distribution plots for each particle set and the final merged set are also presented in Supplementary Fig. S3.

## Model building

The BoNT/X-NTNH/X complex model was built into the map using a combination of automated docking of the individual domains of homologous BoNT/A-NTNH/A complex (PDB code 3v0a)[13] using Phenix[72] and manual building in Coot[73], together with real space refinement in Phenix. Refinement was also carried out in Gromacs using the cryo-EM map as a restraint[74]. The final cryoEM map and model coordinates were deposited into the PDB and EMDB databases under the accession codes 8BYP and EMD-16330, respectively. The cryo-EM data acquisition and model refinement statistics are listed in Supplementary Table 5.

## Ganglioside binding assay

Purified gangliosides GD1a, GD1b, GT1b, and GM1 (Carbosynth, Compton, UK) were dissolved in DMSO and diluted in methanol to reach a final concentration of 2.5 μg/ml; 100 μl were applied to each well of 96-well PVC assay plates (Corning; Corning, NY). After solvent evaporation at 21 °C, the wells were washed with 200 μl PBS/0.1% (w/v) BSA. Nonspecific binding sites were blocked by incubation for 2.5 h at 4 °C in 200 μl of PBS/2% (w/v) BSA. Binding assays were performed in 100 μl PBS/0.1% (w/v) BSA per well for 1 h at 4 °C containing protein samples (in triplicate) at concentrations ranging from 6 μM to 0.003 μM (in serial 2-fold dilution). Samples consisted of the His$_6$-tagged R360A/Y363F inactive mutant of BoNT/X and a His$_6$-tagged H$_C$/A control. Following incubation, wells were washed three times with PBS/0.1% (w/v) BSA and incubated with an HRP-conjugated anti-6xHis monoclonal antibody (1:2000, ThermoFisher) for 1 h at 4 °C. After three washing steps with PBS/0.1% (w/v) BSA, bound samples were detected using Ultra-TMB (100 μl/well, ThermoFisher) as the substrate. The reaction was stopped after 10 min by addition of 100 μl 0.2 M H$_2$SO$_4$, and the absorbance at 450 nm was measured using an Infinite M200PRO plate

reader (Tecan, Männedorf, Switzerland). Data were analyzed with Prism7 (GraphPad Software).

## Determination of the crystal structure of NTNH/X

The NTNH/X protein was obtained as a side-product in the co-purification of the BoNT/X-NTNH/X complex; NTNH/X was expressed in higher amounts than BoNT/X, and the excess pure NTNH/X protein eluted after the BoNT/X-NTNH/X complex during SEC purification. The protein was concentrated to 25 mg/ml and stored at $-80$ °C in 20 mM MES pH 5.5, 150 mM NaCl, 0.5 mM TCEP. NTNH/X was crystallized by the sitting drop vapor diffusion technique in the Swissci 3-well crystallization plates (Swissci, High Wycombe, UK) using the Mosquito crystallization robot (SPT Labtech, Melbourn, UK). The best diffracting crystals were obtained at 15 mg/ml NTNH/X in the storage buffer, in 900 nl drops with a protein:precipitant ratio of 2:1. The reservoir volume was 30 μl and consisted of 27 μl manually mixed Morpheus buffer system 1 and precipitant mix 3 (Molecular Dimensions Limited, Sheffield UK), and 3 μl of Additive Screen condition E3 (Hampton Research, Aliso Viejo, CA, USA)[75]. The final composition of the precipitant solution was 90 mM MES/imidazole pH 6.5, 9% (w/v) PEG 4,000, 18% (v/v) glycerol, and 3% (w/v) dextran sulfate sodium salt. Clusters of elongated hexagon-shaped crystals started growing immediately after set-up and were harvested after 5 weeks and flash-cooled without additional cryoprotection. Diffraction data were collected at the beamline i04 of Diamond Light Source at 100 K and the wavelength of 0.9795 Å. Data collection and refinement statistics are listed in Supplementary Table 7. The diffraction data were processed with DIALS and Aimless[76,77]. The structure was solved by molecular replacement using the program MolRep from the CCP4Interface 8.0.010 with the NTNH/X structure from our cryo-EM model of the BoNT/X-NTNH/X complex as a search model[78,79]. The model was then subjected to iterative cycles of refinement in Refmac 5 in the CCP4Interface 8.0.010 and Phenix version 1.20.1-4487, combined with manual adjustments in Coot 0.9.8.7[79–81]. The refined model was validated with MolProbity and the wwPDB validation server, and had no rotamer or Ramachandran outliers with 92% and 8% of residues lying in the Ramachandran favored and allowed regions, respectively[82,83]. The model coordinates and the electron density maps were deposited into the PDB database under the accession code 8QFT.

## Expression and purification of BoNT/X for stability studies

C-terminally His-tagged inactive BoNT/X construct was expressed in BL2(DE3) *E. coli* cells in TB media induced with 0.5 mM IPTG after the OD$_{600}$ reached 0.8. The cells were harvested after approximately 18 h of post-induction cultivation at 18 °C. Cell pellets were resuspended in lysis buffer (100 mM HEPES, pH 8.0, 500 mM NaCl, 10 mM imidazole, 5% glycerol, 1 mM TCEP), supplemented with DNAse I, lysozyme, and protease inhibitor mix (EDTA-free, EASY-pack, Roche, Basel, Switzerland), and lysed via sonication. The lysed sample was clarified by ultracentrifugation (1 h at 42,000 rpm, 4 °C), bound to a 5 ml HisTrap HP Ni-NTA (Cytiva, Uppsala, Sweden) and eluted with 75 mM imidazole. The protein was further purified by gel filtration using Superdex200 16/60 column (Cytiva, Uppsala, Sweden) pre-equilibrated in 50 mM HEPES, pH 7.2, 500 mM NaCl, 5% glycerol, 1 mM TCEP. The purest fractions containing BoNT/X were collected and concentrated to 3.7 mg/ml.

## Nanoscale differential scanning fluorimetry (nanoDSF)

NanoDSF experiments were carried out in a Prometheus Panta instrument (NanoTemper Technologies, Munich, Germany). Protein samples were diluted to 0.7 mg/ml with buffers used in size-exclusion chromatography experiments (see section "SEC pH stability experiments") and in 50 mM Hepes pH 7.2, 150 mM NaCl, 0.5 mM TCEP. The samples were loaded into the nanoDSF capillaries in triplicates. The ratio between fluorescence at 330 nm and 350 nm which represents the changes in Trp fluorescence intensities was recorded in 20–80 °C temperature range at the ramp rate of

1 °C/min. The ratio and its first derivative, as well as the melting temperature values, were calculated with the manufacturer's software. The nanoDSF data were further analyzed in MoltenProt using an equilibrium thermodynamic model and Gibbs–Helmholtz equation to quantitatively describe protein unfolding and to obtain the Gibbs free energy of unfolding ($\Delta G_u$) extrapolated to 25 °C[33]. NTNH/X unfolding curves were fitted using equilibrium two-state model, which assumes that the protein exists in native (N) or unfolded (U) state with an equilibrium between the folding and unfolding reactions (N ⇆ U). BoNT/X unfolding curves were fitted using equilibrium three-state model, which assumes that the protein exists in three possible states—native (N), unfolded intermediate (I), and unfolded (U) with equilibrium between the states (N ⇆ I ⇆ U), based on the shape of the first-derivative curves. After fitting, values of ($\Delta G_u$) at 25 °C were calculated as the sum of the free energy change for reactions N ⇆ I and I ⇆ U.

## Computational analysis details

Atomistic classical MD simulations were performed to probe the conformational dynamics of isolated BoNT/X and the BoNT/X-NTNH/X complex, constructed based on the experimentally resolved cryo-EM structure of the complex. For the isolated BoNT/X, the BoNT/X subunit was extracted from the BoNT/X-NTNH/X complex, followed by modeling of missing loops in BoNT/X (Met1 and Asn426-Asn465) and NTNH/X (nMet1-nSer8 and nLeu129-nPhe138). The missing loops were modeled in the cryo-EM structure using Modeller10.4 suite, based on a template of the individual subunits generated using AlphaFold2.0[84,85]. Protonation states of titratable residues were predicted based on an electrostatic model using the PropKa3.0 performed for BoNT/X-NTNH/X complex and the individual BoNT/X and NTNH/X subunits (Supplementary Table 9)[86]. Based on these, all residues were modeled in their standard protonation states for the isolated BoNT/X, whilst Glu1048, nGlu554, and nGlu584 were modeled in their protonated form in the complex. The protein models were embedded in a TIP3P water box with 150 mM NaCl. Parameters for the protein, water, and ions were based on the CHARMM36 force-field[87], with a total simulation system size of 206,328 and 297,635 atoms for the BoNT/X and the BoNT/X-NTNH/X complex, respectively. MD simulations were performed in an *NPT* ensemble at $T = 310$ K and $P = 1$ atm, using an integration timestep of 2 fs, and treating long-range electrostatics interaction using the Particle Mesh Ewald (PME) approach. The isolated BoNT/X and the BoNT/X-NTNH/X complex were propagated for 500 ns. In addition, we also performed two replica simulations (each 140 ns) of isolated BoNT/X. All simulations were performed using the NAMD2.14/3.0[88]. Principal component analysis (PCA) on both isolated BoNT/X and BoNT/X-NTNH/X complex was performed using ProDy[89]. Two movies describing the PCA are provided as supporting information (Supplementary Movies 1–2).

## Reporting summary

Further information on research design is available in the Nature Portfolio Reporting Summary linked to this article.

## Data availability

The cryo-EM reconstruction of the BoNT/X-NTNH complex has been deposited in the Electron Microscopy Data Bank under the EMDB accession code EMD-16330 and the atomic model has been deposited in the Protein Data Bank under the PDB accession code 8BYP. The model coordinates and the electron density maps for the crystal structure of NTNH/X were deposited into the PDB database under the accession code 8QFT. Raw data from ganglioside binding assays is provided in Supplementary Data 1.

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

## Acknowledgements

We thank the Swedish National cryo-EM facility staff Marta Carroni, Julian Conrad, and Michael Hall for their help during the acquisition of the cryo-EM datasets. We thank the Diamond Light Source for beamtime (proposal mx15806), and the staff of beamline i04 for assistance with crystal testing and data collection. We thank the National Academic Infrastructure for Supercomputing in Sweden (NAISS 2023/1-31) for computational resources. This work was supported by the Swedish Research Council (2022-03681, 2018-03406) and the Swedish Cancer Society (20 1287 PjF) (P.S.). This work was partially supported by Ipsen. The work of J.S. was supported from ERDF/ESF, OP RDE, project "IOCB Mobility" (No. CZ.02.2.69/0.0/0.0/16_027/0008477) granted to the Institute of Organic Chemistry and Biochemistry of the Czech Academy of Sciences. V.R.I.K acknowledges the Knut and Alice Wallenberg Foundation (V.R.I.K. grant: 2019.0251). A.S. acknowledges the EMBO Long-Term Fellowship (ALTF 952-2022). M.D. was partially supported by grants from the National Institute of Health (NIH) (R01NS080833 and R01NS117626). M.D. holds the Investigator in the Pathogenesis of Infectious Disease award from the Burroughs Wellcome Fund. The data was collected at the Cryo-EM Swedish National Facility funded by the Knut and Alice Wallenberg, Family Erling Persson and Kempe Foundations, SciLifeLab, Stockholm University, and Umeå University.

## Author contributions

P.S. and M.D. conceived the project; J.S. and M.M.-C. carried out the cryo-EM; J.S. carried out the crystallography; A.K. carried out the nanoDSF; A.S and V.R.I.K designed and performed the molecular dynamics simulations and related analysis; J.S. and P.L. carried out the pH stability assays; G.M carried out the ganglioside binding assays, A.K., L.H., J.S., D.B., G.M., and M.E. carried out protein purifications, S.D., F.H., J.P. P.L., and J.Z. carried out cell and in vivo studies; P.S., M.D. and M.B. supervised the project. All authors contributed to the preparation of the manuscript, discussed results, and approved the manuscript. J.S. and M.M.-C. contributed equally to this work.

## Funding

## Competing interests

P.S. and M.D. are inventors on patents regarding BoNT/X. M.B., D.B., M.E., J.P., S.D., and F.H. are employees of Ipsen. The authors declare no additional conflict of interest.
