## [Peer Review File · Communications Chemistry]

This manuscript has been previously reviewed at another *Nature Portfolio* journal. This document only contains reviewer comments and rebuttal letters for versions considered at *Communications Chemistry*.

REVIEWERS' COMMENTS:

Reviewer #1 (Remarks to the Author):

This is an interesting study and I have no further remarks, the authors addressed all the comments.

Reviewer #2 (Remarks to the Author):

The manuscript "Activity of Botulinum neurotoxin X and its structure when shielded by NTN^H" by Martínez-Carranza et al gives an interesting and insightful description of the structural properties of a recently discovered Botulinum neurotoxin: BoNT/X, which display properties different from other botulinum neurotoxins, making it particularly interesting for medical applications.

The manuscript is well written and constructed, and present results of interest for the community. I recommend thus publication.

I have a few comments that should be taken into account by the authors.

Concerning the protonation of residues, on page 10, it is written:

"All residues were predicted to be in their standard protonation states for the isolated BoNT/X, whilst Glu1048, nGlu554, and nGlu584 were predicted to be in their protonated form in the complex."

what is the meaning of nGlu? It is interesting to note that Glu554 and Glu584 are located in the switch region of the translocation domain of BoNT/X which was postulated to interact with membrane of the transmitter vesicle at acidic pH in the article of Lam et al, 2018.

The bibliography format should be revised specially for references 41, 69 which are written as link to pubmed, or reference 40 with a doi, or reference 71 with {\it Phenix} in the title.

Also, reference 41 is written wuth two author names: Rummel & Rummel, whereas there is only one:

Rummel A. Two Feet on the Membrane: Uptake of Clostridial Neurotoxins.

Curr Top Microbiol Immunol. 2017;406:1-37. doi: 10.1007/82_2016_48. PMID: 27921176.

Reviewer #2 (remarks to the author):

Concerning the protonation of residues, on page 10, it is written: "All residues were predicted to be in their standard protonation states for the isolated BoNT/X, whilst Glu1048, nGlu554, and nGlu584 were predicted to be in their protonated form in the complex."

What is the meaning of nGlu? It is interesting to note that Glu554 and Glu584 are located in the switch region of the translocation domain of BoNT/X which was postulated to interact with membrane of the transmitter vesicle at acidic pH in the article of Lam et al, 2018.

Response: This sentence was rephrased as follows to clarify the meaning of nGlu, which follows the convention established in the rest of the article to designate residues belonging to NTNH/X:

'All residues were predicted to be in their standard protonation states for the isolated BoNT/X, whilst in the BoNT/X-NTNH/X complex BoNT/X Glu1048 and NTNH/X nGlu554, nGlu584 were predicted to be in their protonated form.'

*The bibliography format should be revised specially for references 41, 69 which are written as link to pubmed, or reference 40 with a doi, or reference 71 with *{\it Phenix}* in the title. Also, reference 41 is written wuth two author names: Rummel & Rummel, whereas there is only one:*

Rummel A. Two Feet on the Membrane: Uptake of Clostridial Neurotoxins. Curr Top Microbiol Immunol. 2017;406:1-37. doi: 10.1007/82_2016_48. PMID: 27921176.

Response: the reference entries have been inspected to correct all errors of the kind pointed by reviewer #2.